# Novel Iron Chelators, Super-Polyphenols, Show Antimicrobial Effects against Cariogenic *Streptococcus mutans*

**DOI:** 10.3390/antibiotics12111562

**Published:** 2023-10-24

**Authors:** Yuki Shinoda-Ito, Kazuhiro Omori, Takashi Ito, Masaaki Nakayama, Atsushi Ikeda, Masahiro Ito, Toshiaki Ohara, Shogo Takashiba

**Affiliations:** 1Department of Pathophysiology-Periodontal Science, Faculty of Medicine, Dentistry and Pharmaceutical Sciences, Okayama University, 2-5-1 Shikata-cho, Kita-ku, Okayama 700-8525, Japanstakashi@okayama-u.ac.jp (S.T.); 2Center for Innovative Clinical Medicine, Okayama University Hospital, 2-5-1 Shikata-cho, Kita-ku, Okayama 700-8558, Japan; 3Department of Oral Microbiology, Faculty of Medicine, Dentistry and Pharmaceutical Sciences, Okayama University, 2-5-1 Shikata-cho, Kita-ku, Okayama 700-8525, Japan; 4Department of Periodontics & Endodontics, Division of Dentistry, Okayama University Hospital, 2-5-1 Shikata-cho, Kita-ku, Okayama 700-8558, Japan; aikeda.0429@okayama-u.ac.jp; 5Department of Pathology and Experimental Medicine, Faculty of Medicine, Dentistry and Pharmaceutical Sciences, Okayama University, 2-5-1 Shikata-cho, Kita-ku, Okayama 700-8558, Japan; t_ohara@cc.okayama-u.ac.jp

**Keywords:** antimicrobial, iron chelator, oral infection, *Streptococcus mutans*, super-polyphenols

## Abstract

Dental caries are an oral infectious disease that can affect human health both orally and systemically. It remains an urgent issue to establish a novel antibacterial method to prevent oral infection for a healthy life expectancy. The aim of this study was to evaluate the inhibitory effects of novel iron chelators, super-polyphenols (SPs), on the cariogenic bacterium *Streptococcus mutans*, in vitro. SPs were developed to reduce the side effects of iron chelation therapy and were either water-soluble or insoluble depending on their isoforms. We found that SP6 and SP10 inhibited bacterial growth equivalent to povidone-iodine, and viability tests indicated that their effects were bacteriostatic. These results suggest that SP6 and SP10 have the potential to control oral bacterial infections such as *Streptococcus mutans*.

## 1. Introduction

Dental caries are an infectious disease caused by the formation of oral biofilms, known as dental plaque, in which a variety of pathogenic and non-pathogenic bacteria accumulate [1,2,3]. The causative pathogens are also associated with systemic diseases such as diabetes, aspiration syndrome, pneumonia, and endocarditis [4,5]. In addition, the number of elderly people in Japan who cannot brush their own teeth is increasing. Unfortunately, mechanical removal such as brushing and scaling is still the main method for removing dental biofilm and maintaining oral hygiene. Therefore, there is an urgent need to establish a novel method to control oral bacterial infections to maintain a healthy life expectancy and to reduce the burden on medical personnel [6]. In addition, the easy use of antibiotics against bacterial infections has led to the spread of antimicrobial resistance (AMR), which has become a global problem [7,8]. Therefore, there is a need to develop new concepts and materials to control pathogenic bacterial infections as an alternative to antimicrobial agents.

Iron is an essential minor metal for organisms and is involved in various biological reactions such as hemoglobin synthesis, oxidation-reduction reactions, cell growth and apoptosis. Bacteria also require iron to survive [9]. It has been reported that bacterial growth can be inhibited by iron deficiency [10]; however, the majority of iron in organisms exists in conjugate or heme forms, leaving very few iron molecules available for bacterial uptake [11]. In addition, the human body has a biopurification system to reduce free iron in response to bacterial invasion [12,13]. To counter this host system, some microbial strains can produce iron-binding small molecules called siderophores for efficient iron uptake [14]. Consequently, as a countermeasure against bacteria, hosts defend themselves by binding the neutrophil gelatinase-associated lipocalin to siderophores [15]. Thus, there is a constant interaction between host and bacteria, as the latter need to obtain iron for their survival.

Some reports link oral microbes to iron. One study suggests that the addition of iron (3.6 μM) has been reported to increase the level of the steady-state growth of the cariogenic organism *Streptococcus mutans* OMZ176 2.8-fold [16]. Another study suggests that the hemolysin produced by the periodontopathogenic organism *Porphyromonas gingivalis* (*P. gingivalis*) is involved in providing hemoglobin to periodontal pockets [17]. In addition, some studies suggest that deferoxamine (DFO) and deferasirox (DFX), representatives of commercial iron chelators, inhibit the growth of periodontopathic bacteria [18,19]. However, these iron chelators cause severe side effects in the host such as diarrhea and emesis, making them difficult to consider for use in the control of oral bacterial infections [20,21].

The super-polyphenols (SPs) that have been developed by Disease Absorption System Technologies (DASTec) Co., Ltd. (Kanazawa, Japan) are novel iron chelators for use in iron chelation therapy. These compounds are synthesized from various functional groups that exhibit chelating activity and from polymers such as chitosan, glucosamine, an amino sugar, or histidine, which are amino acids. Recently, Ohara et al. reported that the water-soluble SP6 and SP10 had the strongest chelating abilities and anti-tumor effects without side effects [22]. Based on these results, we hypothesized that SPs could be used to treat oral infections. This study focused on these novel chemical agents, SPs, and investigated the efficacy of SPs on *Streptococcus mutans*.

## 2. Materials and Methods

### 2.1. Reagents

SPs (SP1, 5, 6, 9, 10; Figure 1) were provided by DASTec Co., Ltd. (Kanazawa, Japan). The SPs were chemically synthesized from catechol and benzoic acids. SP1 was made by a condensation reaction with chitosan. SP5 and SP6 were made by a condensation reaction with glucosamine, and SP9 and SP10 were made by a condensation reaction with histidine. The SPs were developed to have few side effects by avoiding metabolism by cytochrome P450 [22]. The SPs were dissolved to prepare a stock solution at a concentration of 10 mg/mL in sterile distilled water and used at various concentrations ranging from 100 to 1000 µg/mL in the respective culture media. Povidone-iodine (10%; Negmin^®^, Mylan, Tokyo, Japan) was diluted to 1% (the clinical dose used for a gargle) and used as a positive control. Approximately 100 µL/mL of sterile distilled water was used as a negative control because it corresponds to the maximum concentration of SPs at the concentration of 1000 µg/mL.

### 2.2. Bacteria

*Streptococcus mutans* ATCC 25175 (*S. mutans*) was cultured in tryptic soy broth with yeast extract (TSBY; Becton, Dickinson and Company, Sparks, MD, USA) until the bacterial logarithmic growth phase. Tryptic soy broth without dextrose with yeast extract (Becton, Dickinson, and Company) and 1% sucrose (Nakalai Tesque, Kyoto, Japan) (TSBY-s) were used for biofilm studies. The bacterial suspensions were diluted to 1 × 10^7^ or 1 × 10^9^ colony forming units (CFU)/mL with each medium by determining the absorbance at 660 nm using a photometer (Miniphoto 518R; Taitec, Saitama, Japan) for subsequent examinations, and incubated at 37 °C.

### 2.3. Human Cells

Human gingival keratinocytes (HGKs) purchased from American Type Culture Collection (ATCC, Manassas, VA, USA) were used in this study and cultured in a Dermal Cell Basal Medium (ATCC) containing a Keratinocyte Growth Kit (ATCC).

### 2.4. Bacterial Growth Assay

Bacterial turbidity was measured to identify the SPs that had antibacterial effects on oral bacteria. Approximately 1 × 10^7^ CFU/mL of *S. mutans* was incubated with the SP solutions (SP1, 5, 6, 9, 10; 100, 300, 500, and 1000 µg/mL) in 96-well microtiter plates (Corning, New York, NY, USA) at 37 °C for 18 h. Absorbance was measured at a wavelength of 595 nm. Based on the results of the bacterial growth assay, SP6 and SP10 were chosen for further experiments.

### 2.5. Bacterial Morphological Observation

Morphological changes in *S. mutans* were observed by microscopy after Gram staining (ScyTek Laboratories, Logan, UT, USA) and by scanning electron microscopy (SEM) after percolation. *S. mutans* (1 × 10^9^ CFU/mL) was incubated in an SP (SP6 and SP10; 1000 µg/mL) solution at 37 °C for 1 h. For the observation of the Gram-stained bacteria, a 5-µL aliquot of the bacterial suspension was dropped and flame-fixed onto microscope glass slides. The glass slides were processed according to the manufacturer’s instructions, and observed by microscopy (Olympus BX50; Olympus, Tokyo, Japan). For SEM observation, a 1-mL aliquot of the bacterial suspension was percolated using a nano-percolator (JEOL, Tokyo, Japan). Then, the filtered bacteria were immobilized by 50–99.9% graded ethanol. Bacteria fixed on filters were attached to the SEM sample stage, coated with osmium in vacuum and observed by SEM (S-4800; Hitachi High-Tech Corporation, Tokyo, Japan) under the voltage of 15 kV.

### 2.6. Bacterial Viability Analysis

The LIVE/DEAD^®^ BacLight™ Bacterial Viability Kit (Life Technologies, Waltham, MA, USA) was used to determine the bacterial viability in the SP (SP6 and SP10; 100 and 1000 µg/mL, respectively) solutions. Bacterial viability was evaluated by the fluorescence quantification of the stained bacteria using a fluorescence microplate reader and by direct observation of the stained bacteria using a fluorescence microscope. For fluorescence quantification, *S. mutans* (1 × 10^7^ CFU/mL) in SP (SP6 and SP10; 1000 µg/mL) solutions was incubated in 96-well microplates at 37 °C for up to 18 h. Subsequent staining of bacteria was performed according to the manufacturer’s protocol. At 1 h and 18 h after incubation, the fluorescence intensity of the stained samples was measured at 530 nm and 630 nm using a fluorophotometer (Gemini XPS, San Jose, CA, USA) at an excitation wavelength of 485 nm. The ratio of fluorescence observed at 530 nm to the fluorescence at 630 nm was measured to determine the ratio of the number viable cells (SYTO^®^9 positive cells) to the number of dead cells (propidium iodide positive cells). For the direct observation of bacteria by fluorescence microscopy, *S. mutans* (1 × 10^9^ CFU/mL) was incubated in SP (SP6 and SP10; 1000 µg/mL) solutions at 37 °C for 1 h. The subsequent procedures for bacterial staining were performed according to the manufacturer’s protocol. A 5-µL aliquot of the bacterial suspension was dropped onto glass slides, covered by coverslips, and finally observed under a fluorescence microscope (Olympus BX50, Tokyo, Japan).

### 2.7. Biofilm Forming Assay

Crystal violet staining was performed to investigate the effect of SP6 and SP10 (100, 300, 500, and 1000 µg/mL, respectively) on biofilm formation. *S. mutans* (1 × 10^7^ CFU/mL) was incubated in SP solution in 96-well plates at 37 °C for 18 h. The incubated plates were washed twice with phosphate buffered saline (PBS; Thermo Fisher Scientific, Waltham, MA, USA), fixed with 99.8% methanol, and stained with 0.1% crystal violet (Sigma-Aldrich, St. Louis, MO, USA)/PBS solution. The plates were carefully washed with distilled water until the excess stain was removed, and the remaining stain was eluted with 99.5% ethanol and transferred to new plates. Absorbance was measured at a wavelength of 595 nm (SH-1000; Corona Electric Co., Ibaraki, Japan).

### 2.8. Bacterial Iron Uptake Assay

Iron uptake by *S. mutans* in the presence of SP6 and SP10 (100 and 1000 µg/mL, respectively) was quantified using the fluorescent reagent for the detection of bivalent iron in cells, Ferro Orange (Dojindo Laboratories, Kumamoto, Japan). Approximately 1 × 10^7^ CFU/mL of *S. mutans* was incubated in an SP solution at 37 °C for 1 or 18 h in tubes. The bacterial suspension was centrifuged at 3000 revolutions per minute (rpm) for 10 min, and the supernatant was discarded. It was then washed and resuspended three times in PBS. A stock solution of Ferro Orange was dissolved in dimethyl sulfoxide (Sigma-Aldrich) according to the manufacturer’s brochure. The Ferro Orange stock solution dissolved in PBS was added to the centrifuged bacteria, and the resuspended solution was transferred to 96-well microplates and incubated for 30 min at room temperature under light-shielded conditions. The fluorescence intensity of the stained samples was measured at 590 nm using a fluorophotometer (Gemini XPS; Molecular Devices, Sunnyvale, CA, USA) at an excitation wavelength of 544 nm.

### 2.9. Cell Toxicity Test on Human Cells

Cytotoxicity to HGKs was assessed using a reagent containing a tetrazolium compound, Cell Counting Kit-8 (Dojindo, Mashiki, Japan). HGKs were seeded at 1 × 10^4^ cells/well on 96-well microplate and incubated for 24 h to ensure that they adhered to the bottom of the plates. After 24 h of incubation in a medium containing SPs, the plates were washed three times by PBS, and 100 µL of medium was added into the wells. Color development was observed according to the manufacturer’s technical brochures. Absorbance was measured at a wavelength of 450 nm (SH-1000).

### 2.10. Statistical Analysis

All results were confirmed by at least three independent experiments, each of which was performed in triplicate. Data are presented as the mean ± standard deviation (SD) from at least three independent experiments. One-way or two-way analysis of variance (ANOVA) was used to test the differences between three or more groups, and the Tukey–Kramer test was used for multiple comparisons. Statistical analysis was performed using Prism 8 for Windows (version-8.4.3: GraphPad Software, San Diego, CA, USA), and a *p*-value (*p*) of less than 0.05 indicated statistical significance.

## 3. Results

### 3.1. SPs Suppresses Bacterial Growth

The bacterial growth of *S. mutans* was assessed by measuring turbidity. To determine which SPs could inhibit bacterial growth, different types of SPs (SP1, 5, 6, 9, 10) were tested. It was found that SP5 at over 1000 µg/mL, SP6 and SP10 at over 300 µg/mL were effective in inhibiting the bacterial growth of *S. mutans* (Figure 2), compared to the solution without SPs. SP6 and SP10 have the same chelating structural parts (six-membered carbon ring with three hydroxy groups) but different basic formulas (SP6: glucosamine, SP10: histidine; Figure 1). Based on the above results that SP6 and SP10 were particularly effective in suppressing bacterial growth, subsequent experiments focused on SP6 and SP10.

### 3.2. SP6 and SP10 Did Not Affect Bacterial Morphology

The bacterial morphology of *S. mutans* after treatment with SP6 and SP10 at 37 °C for 1 h was observed microscopically after Gram staining or by SEM after filtration. The bacterial chains were preserved under all conditions in the Gram-stained images (Figure 3A). Observation by SEM revealed that although there was clear adhesion of chemical substances around the bacteria immersed in SP6 and SP10, no apparent changes in the bacterial morphology were shown compared to the negative controls (no SPs). On the other hand, the bacteria immersed in the povidone-iodine solution did not maintain their normal morphology (Figure 3B).

### 3.3. SP6 and SP10 Did Not affect Bacterial Viability

The bacterial viability of *S. mutans* was assessed by morphological observation and the quantification of fluorescence intensity after treatment with SP6 and SP10 at 37 °C for 1 or 18 h and LIVE/DEAD^®^ staining (Figure 4). When quantifying the bacterial survival rate by fluorescence intensity, there was no significant difference between any of the conditions at 1 h after immersion in SP6 and SP10 solutions. At 18 h, the bacterial survival rate was significantly lower only in the povidone-iodine group (positive control) compared to the no SPs group (negative control), while bacteria in the SP6 and SP10 groups tended to survive (Figure 4A). The morphological observation of the bacteria performed 1 h after the addition of SP6 and SP10 also showed the same trend as that of the fluorescence intensity measurement results. Specifically, the bacteria were alive in the no drug group and even when either SP6 or SP10 was added at high concentrations. In contrast to the SP solutions, the bacteria in the povidone-iodine solution were damaged (Figure 4B).

### 3.4. SP6 and SP10 Suppresses Biofilm Formation

Since the virulence of *S. mutans* is enhanced by the formation of biofilms, it is necessary to investigate whether SPs have an effect on biofilm formation as well as bacterial growth suppression. The inhibitory effect of SP6 and SP10 on the *S. mutans* biofilm formation was evaluated by a colorimetric method. It was observed that over 300 µg/mL of SP6 and over 500 µg/mL of SP10 significantly inhibited biofilm formation (Figure 5).

### 3.5. SP6 and SP10 Suppresses Bacterial Iron Uptake

Iron uptake by *S. mutans* in the presence of SP6 and SP10 was examined using a fluorescent reagent to detect the bivalent iron. *S. mutans* in 1000 µg/mL of SP6 and SP10 tended to take up less iron than that in TSBY alone, especially after 18 h incubation. Povidone-iodine also inhibited iron uptake by bacteria, but the decrease was less than that of the SPs (Figure 6).

### 3.6. SP6 and SP10 Was as Cytotoxic as Povidone-Iodine

The cytotoxicity of SPs on HGKs was evaluated using a colorimetric method with the tetrazolium compound. Over 300 µg/mL of SP6 and SP10 showed significant cytotoxicity as well as the positive control, 1% povidone-iodine (Figure 7).

## 4. Discussion

The frequent and excessive use of antibiotics over the last several decades has led to rampant antibiotic resistance in bacteria [7,8]. The rise and spread of AMR is currently a major global threat. If this situation continues, the number of deaths due to AMR is expected to increase dramatically, resulting in significant economic losses. In response to this major social problem, action plans against AMR have been developed in many countries around the world, and the promotion of research and development of new antimicrobial agents effective against AMR organisms has been taken up as an important issue. However, in recent years, many pharmaceutical companies, including those in Japan, have withdrawn from the research and development of new antimicrobial agents due to difficulties in business feasibility, drug discovery research and clinical development, and the development pipeline has been depleted worldwide. However, the control of bacterial infection by targeting iron metabolism, which was the focus of this study, proposes an approach from an unprecedented perspective [23]. Therefore, it is important to investigate the efficacy of SPs as a potential new alternative to existing antimicrobial agents.

Bactericides such as povidone-iodine and cetylpyridinium chloride in mouthwashes and toothpastes are effective against oral infections; however, these chemical agents sometimes cause the bacterial flora to grow uncontrollably [24]. Therefore, bacteriostatic agents would be ideal materials to control oral pathogenic bacteria without disrupting the oral microbiota. SPs including SP6 and SP10 are biologically safe agents that have been developed to produce fewer side effects in iron chelation therapy. SP6 and SP10 (600 and 1000 mg/kg) caused no acute and basic toxicity, no body weight change, no abnormal behaviors, and no tissue damage (kidney and liver) during examination in rats when orally administered. A blood test also demonstrated no significant abnormalities. In addition, to compare the safety of SP6 and SP10 to the commercially available iron chelator DFO, an intravenous injection test showed that all mice died immediately after intravenous administration of DFO (300 mg/kg), on the other hand, none died after the administration of SP6 or SP10 (300 mg/kg). These results suggest that SP6 and SP10 are basically safer than DFO [22]. Recent systematic review has also reported that DFO is a first line for iron overload that can sometimes cause kidney disease; especially, a proximal tubulopathy pattern may be observed in treatment with DFO [25]. Therefore, the safety of SPs as a new iron chelator to replace DFO is believed to be very high, and SPs are easy to use in their application as oral bacterial infection control agents.

Povidone-iodine, a complex of polyvinylpyrrolidone and iodine, is commonly used in mouthwashes and disinfectants to infiltrate the oral biofilm and bacterial cell walls [26]. In this study, the SPs were investigated to determine which SPs exhibited antibacterial effects on planktonic oral bacteria. The results showed that the bacterial turbidity of *S. mutans* in the presence of SP6 and SP10 (>300 µg/mL) was significantly reduced to the same level as povidone-iodine. Specifically, SP6 and SP10 have the same chelating structure, with three hydroxy groups on benzene, and have been shown to inhibit the bacterial growth and biofilm formation of *S. mutans* at relatively low concentrations compared to other SPs (Figure 2 and Figure 5). In recent years, the use of iron chelators in the treatment of infectious diseases has been reported. Bereswill et al. has reported recently that the disease-alleviating effects of specific iron binding by oral DFO application in the infection of *Campylobacter jejuni* induced acute enterocolitis in mouse models [27]. The antimicrobial and anti-biofilm activity of DFO against the oral bacteria *Prevotella intermedia* and *P. gingivalis* has also been reported [18,19]. These results suggest that iron chelation capacity may be important for antimicrobial activity. And it is suggested that SP6 and SP10 are compounds that can be used in antimicrobial therapy as new iron chelators. However, further investigation is needed to determine why the strength of iron chelating capacity affects antimicrobial activity. In addition, SP6 and SP10 were found to be as cytotoxic as 1% povidone-iodine in an in vitro toxicity test on human-derived cells (Figure 7). Considering that 1% povidone-iodine is a concentration that has already been used clinically in dentistry, further studies on its clinical safety would also be necessary.

SPs are new compound, and their efficacy has been examined mainly in the field of cancer oncology. In particular, it has recently been reported that the administration of DFX and SP10, the compound used in this study, can regulate iron metabolism and inhibit the growth of poor prognosis esophageal cancer cells and triple-negative breast cancer cells [28,29]. Thus, iron metabolism control by iron chelation is attracting attention as a new cancer treatment strategy. In addition, the effect of iron chelation using SPs may be a target for further attention in the development of a new therapeutic strategy for AMR.

A limitation of this study is that it was only able to examine the effects of the antimicrobial activity of SPs against single cultures of bacteria (*S. mutans*) associated with oral infections. It is essential to investigate the effects of SPs on oral and intestinal microflora in an in vivo model in comparison to commercially available iron chelators, including DFO or DFX.

## 5. Conclusions

This in vitro study showed that novel iron chelators, SP6 and SP10, have a bacteriostatic (growth inhibiting) effect on *S. mutans* by inhibiting iron uptake. SP is expected to be a novel drug with fewer side effects and better antimicrobial efficacy than DFO and DFX. To determine the future clinical applications of SP6 and SP10 to oral bacterial infections, their antibacterial effects on other oral microbes and biological safety should be comprehensively analyzed.

## Figures and Tables

**Figure 1 antibiotics-12-01562-f001:**
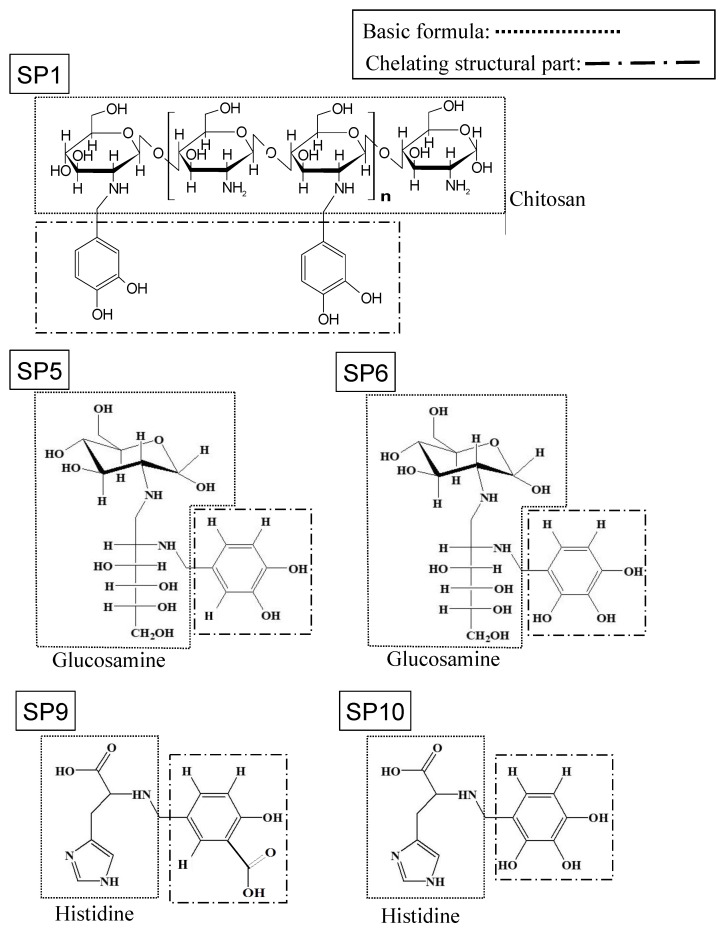
Structural formulas of SPs. The structural formula of SPs: basic formulas (chitosan, glucosamine, or histidine) and chelating structural moieties (dihydroxybenzenes, trihydroxybenzenes, or salicylic acid) were combined. Each SP was synthesized from the following materials: SP1, chitosan with dihydroxybenzenes; SP5, glucosamine with dihydroxybenzenes; SP6, glucosamine with trihydroxybenzenes; SP9, histidine with salicylic acid; and SP10, histidine with trihydroxybenzenes.

**Figure 2 antibiotics-12-01562-f002:**
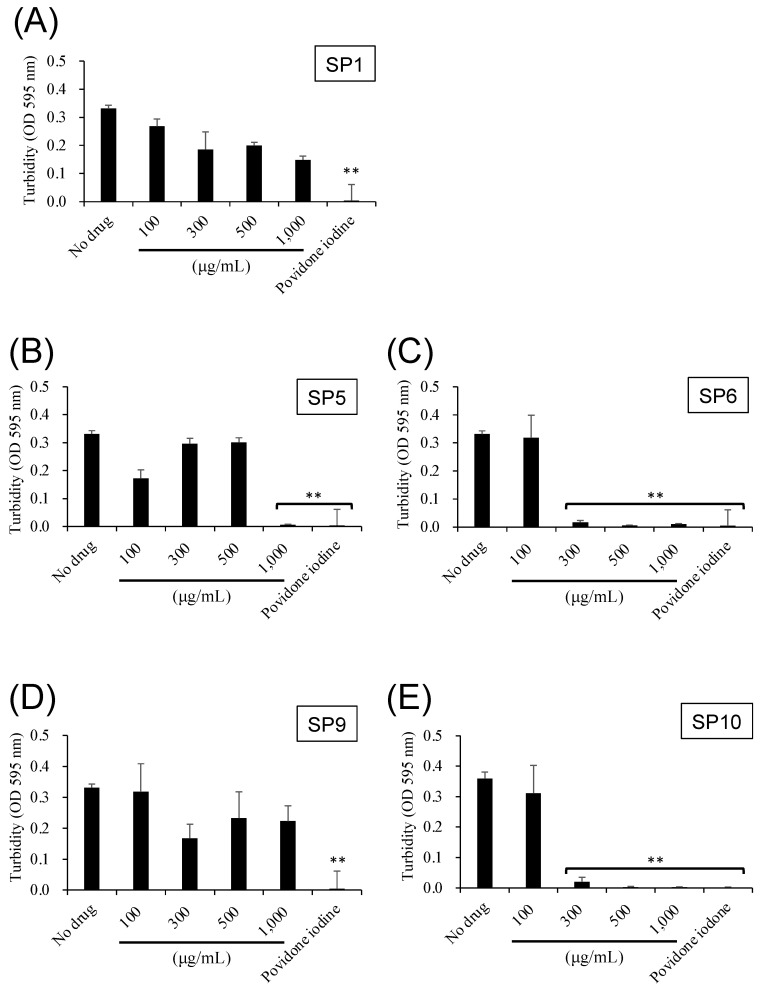
Antibacterial effects of SPs on bacterial growth. The *y*-axis shows the turbidity of *S. mutans* (OD 595 nm). The *x*-axis shows the concentrations of SP1, 5, 6, 9, 10 (100, 300, 500, and 1000 µg/mL). 1% povidone-iodine stimulation was used as a positive control. *n* = 3, ** *p* < 0.001 (vs. no drug). (**A**) SP1, (**B**) SP5, (**C**) SP6, (**D**) SP9, (**E**) SP10.

**Figure 3 antibiotics-12-01562-f003:**
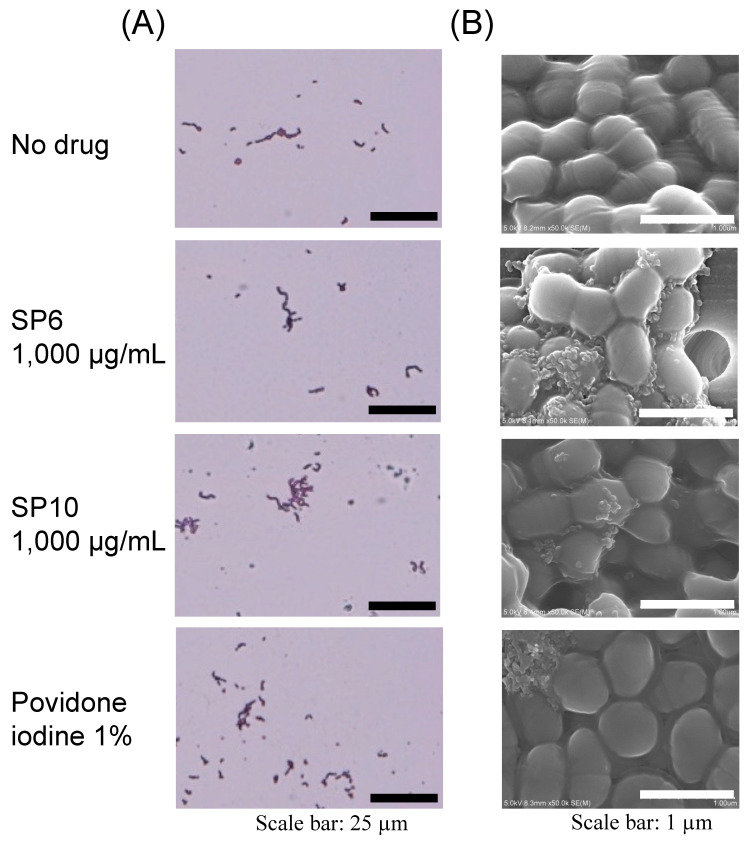
Effect of SP6 and SP10 on bacterial morphology. (**A**) Gram-stained bacterial morphology of *S. mutans* after incubation in SPs solution for 1 h observed by microscopy. 1% povidone-iodine was used as a positive control. Three independent experiments were performed, and typical images are shown. Scale bar: 25 µm. (**B**) Bacterial morphology of *S. mutans* after incubation in SPs solution for 1 h and filtration observed by SEM. 1% povidone-iodine was used as a positive control. Three independent experiments were performed, and typical images are shown. Scale bar: 1 µm.

**Figure 4 antibiotics-12-01562-f004:**
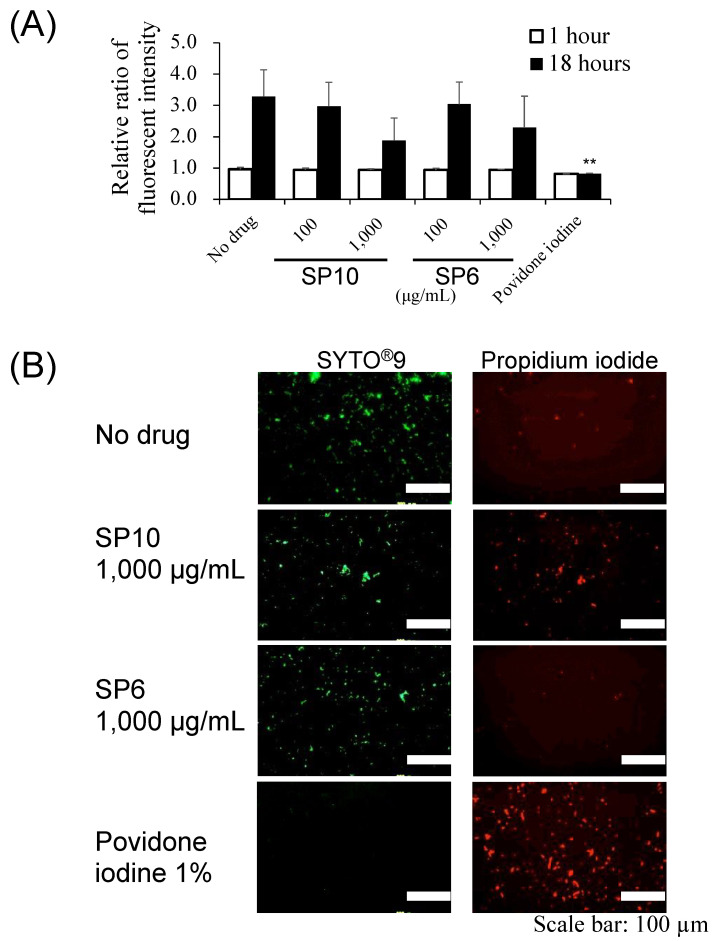
Effect of SP6 and SP10 on bacterial viability. (**A**) *S. mutans* in the SPs was stained with LIVE/DEAD^®^ staining reagent after incubation in SP solutions for 1 or 18 h, and the fluorescence intensity of stained bacteria was measured at 530 nm and 630 nm using a fluorophotometer (Gemini XPS). 1% povidone-iodine was used as a positive control. The *y*-axis shows the ratio of the number viable cells (SYTO^®^9 positive cells) to the number of dead cells (propidium iodide positive cells). *n* = 3, ** *p* < 0.001 (vs. no drug of 18 h). (**B**) *S. mutans* was stained with LIVE/DEAD^®^ staining reagent after incubation in SPs solution for 1 h, and cell morphology was observed under a fluorescence microscope. 1% povidone-iodine was used as a positive control. Three independent experiments were performed, and representative images are shown. Scale bar: 100 µm.

**Figure 5 antibiotics-12-01562-f005:**
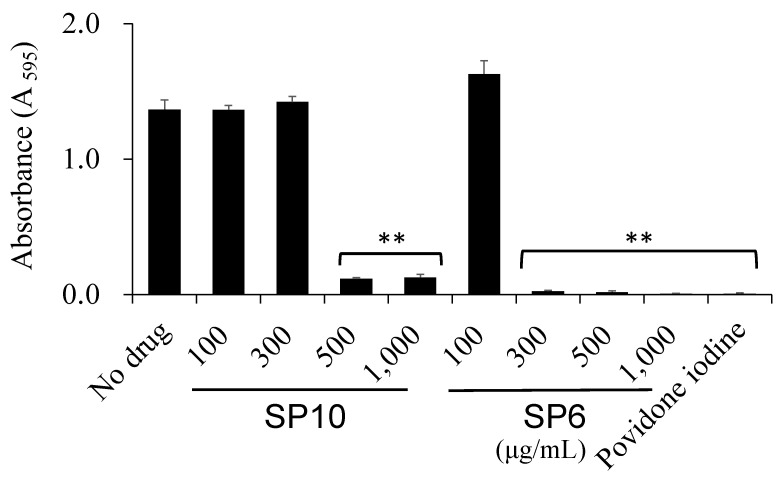
Antibacterial effects of SP6 and SP10 on biofilm formation. *S. mutans* biofilm was formed by incubating the bacteria in sucrose-added TSBY for 18 h. The biofilm was stained with crystal violet, and the absorbance of the eluted dye was measured. Stimulation with 1% povidone-iodine was used as a positive control. The *x*-axis shows the concentration of SP6 and 10 (100, 300, 500, and 1000 µg/mL). The *y*-axis shows the absorbance of the eluted dye from the stained biofilm at 595 nm (A _595_). *n* = 3, ** *p* < 0.001 (vs. no drug).

**Figure 6 antibiotics-12-01562-f006:**
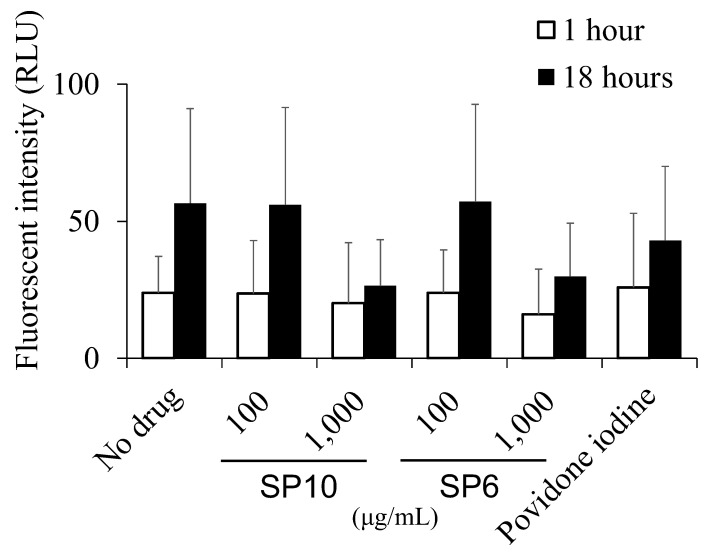
Effect of SP6 and SP10 on bacterial iron uptake. Iron uptake by *S. mutans* in SPs was examined based on the fluorescence intensity of the stained bacteria. 1% povidone-iodine was used as a positive control. The *x*-axis shows the concentration of SP6 and 10 (100 and 1000 µg/mL). The *y*-axis shows the fluorescence intensity of the stained bacteria at a wavelength of 590 nm. *n* = 3.

**Figure 7 antibiotics-12-01562-f007:**
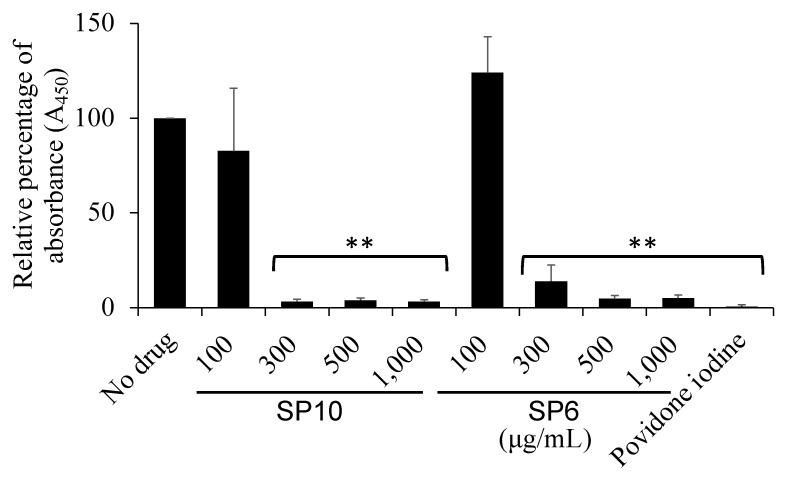
Cytotoxicity of SP6 and SP10 on human cells. The cytotoxicity test on HGKs was performed using a colorimetric method with the tetrazolium compound. 1% povidone-iodine was used as a positive control. The *x*-axis shows the concentration of SP6 and SP10 (100, 300, 500, and 1000 µg/mL). The *y*-axis shows the absorbance measured at 450 nm. *n* = 3, ** *p* < 0.001 (vs. no drug).

## Data Availability

The data that support this study are available from the corresponding author upon reasonable request.

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
