# Peer review of "Novel Iron Chelators, Super-Polyphenols, Show Antimicrobial Effects against Cariogenic Streptococcus mutans"

_antibiotics, 2023, doi:10.3390/antibiotics12111562_

Round 1

Reviewer 1 Report

Comments and Suggestions for Authors

I have gone through the manuscript ID: antibiotics-2631489 entitled: Novel iron chelators, super-polyphenols, show antimicrobial effects against cariogenic Streptococcus mutans. Authors describe new super polyphenols capable of inhibiting the growth of cariogenic S. mutans. Even though the research idea is interesting, some crucial and important information in the manuscript is missing. My main concerns refer to the design of the experiments and the lack of discussion. After a major revision the manuscript could be considered for publication. I have listed my comments below.

INTRODUCTION

Certain sentences are not clearly written. Please reformulate it.

Sentence “As a result, the growing elderly population in Japan has consequently……” it is not clear as a result of what? Please write it more clearly

Sentence: “Super-polyphenols (SPs) have been developed by Disease Absorption System Technologies (DASTec) Co., Ltd to reduce the side effects of iron chelation therapy.”…Do you mean that they are designed in order to reduce harmful effect of commercial iron chelators like deferoxamine? It is not clearly written. Are there any studies showing non-harmful effect of this super-polyphenols given that you refer to their iron chelation activity as a mechanism of antibacterial activity

MATERIALS AND METHODS

Reagents subsection: please give more detailed description about SPs and their structures. Only information about this were given in supplementary.

Bacterial growth assay subsection: give information about concentrations of SP used.

Biofilm forming assay subsection: give information about the number of CFU/ml of S. mutans that was used for the experiment. Also, the concentrations of SPs used are missing. In this part it says that the effect of SPs was investigated, but only SP6 and SP10 were examined, like in all other assays. It is necessary to harmonize the writing of methods. It can be written at the beginning of methods description that based on the results of bacterial growth assay only SP6 and SP10 were chosen for further investigation. Also, is it possible to present the results of biofilm assay as a % of biofilm biomass, that way you have a clearer picture of results obtain, and you can discus about inhibition of biofilm formation by SPs.

Bacterial iron uptake assay subsection: give information about concentrations of SP used. Please explain way was the iron uptake followed for 1h and 18h.

Bacterial morphological observation subsection: give information about SPs concentration used.

Bacterial viability analysis subsection: This whole part of methodology in not clearly written. Reformulate in order to make sense.

 RESULTS

Presentation of results is not good. Please explain why was only chitosan investigated in first experiment (inhibition of bacterial growth). If you wanted to test basic formulas of SPs, then why did you test histidine and glucosamine as well, and in all experiments.

Is it necessary to test H2O? Even though SPs are dissolved in water, in my opinion it is not necessary to test it as solvent control. In case you want water to stay, then all statistic analysis must be done compared to solvent control not to no drug. And solvent control must be compared in that case to no drug.

The concentration selection for testing bacterial iron uptake as well as viability is questionable. You stated that concentration of 1000 µg/ml was used, but at that concentration based on the results of bacterial growth inhibition assay no growth was detected. So, if there are no bacteria it is not possible to test effect of SPs on their iron uptake. Furthermore, how did you test the ability of SPs to chelate iron? Whole research is based on the ability of these compounds to competitively bind iron ions and consequently reduce ability od bacteria to chelate iron that can lead to their death.

It would be good to present results of cytotoxicity as % of cell survival, not as absorbance.

 DISCUSSION

Discussion must be more detailed, explaining obtained results. In present form, in discussion part, results are just repeated, without any specific explanation. This whole part must be expanded and written again.  

Comments on the Quality of English Language

Overall quality of English Language is good. Needs minor editing. 

Author Response

Reviewer 1
Comments and Suggestions for Authors
I have gone through the manuscript ID: antibiotics-2631489 entitled: Novel iron chelators, super- polyphenols, show antimicrobial effects against cariogenic Streptococcus mutans. Authors describe new super polyphenols capable of inhibiting the growth of cariogenic S. mutans. Even though the research idea is interesting, some crucial and important information in the manuscript is missing. My main concerns refer to the design of the experiments and the lack of discussion. After a major revision the manuscript could be considered for publication. I have listed my comments below.

Thank you for reviewing the manuscript and providing several useful comments. We have provided point- by-point responses to each comment, and the red text indicates the revised manuscript.

INTRODUCTION
Certain sentences are not clearly written. Please reformulate it.
Sentence “As a result, the growing elderly population in Japan has consequently......” it is not clear as a result of what? Please write it more clearly
Sentence: “Super-polyphenols (SPs) have been developed by Disease Absorption System Technologies (DASTec) Co., Ltd to reduce the side effects of iron chelation therapy.”...Do you mean that they are designed in order to reduce harmful effect of commercial iron chelators like deferoxamine? It is not clearly written. Are there any studies showing non-harmful effect of this super-polyphenols given that you refer to their iron chelation activity as a mechanism of antibacterial activity.

We have revised the text based on the reviewer’s suggestion. Co-author Ohara has reported previously that Super-polyphenol has lower biohazard effects than existing drugs (deferasirox; DFO) while maintaining high iron chelating capacity (ref. 22). In this paper, Ohara examined SPs basic chelating abilities using standard ferric (Fe3+) water solution by the sulfosalicylic acid visual colorimetric method. SP6 and SP10 chelated Fe3+ in a dose-dependent manner. A dose of 0.5 mg/mL SP6 chelated more than 75% of iron, and 0.1 mg/mL SP10 chelated more than 80% of iron. In addition, acute toxicity tests in rats were performed to evaluate the basic toxicity of SP6 and SP10. SPs were orally administered (600 and 1,000 mg/kg), and the body weight of the rats did not change compared to the control group during the examination. No abnormal behaviors were observed through the period of examination. A blood test was also performed in rats treated with SP6 (1,000 mg/kg) and SP10 (1,000 mg/kg), and no significant abnormalities were observed. To evaluate any possible adverse effects to the organs, Ohara examined the liver and kidney tissues and found no injuries in the specimens. To compare the safety of SP6 and SP10 to the known iron chelator DFO, an intravenous injection test was performed. Although all mice died immediately after intravenous administration of DFO (300 mg/kg), none died after administration of SP6 or SP10 (300 mg/kg). We think these results demonstrated that SP6 and SP10 are basically safer than DFO.

MATERIALS AND METHODS

Reagents subsection: please give more detailed description about SPs and their structures. Only information about this were given in supplementary.

We have revised the text based on the reviewer’s suggestion about SPs and their structures in Figure 1.

Bacterial growth assay subsection: give information about concentrations of SP used.

We have revised the text based on the reviewer’s suggestion about concentrations of SP solutions (SP1, 5, 6, 9, 10; 100, 300, 500, and 1,000 μg/mL) used in bacterial growth assay.

Biofilm forming assay subsection: give information about the number of CFU/ml of S. mutans that was used for the experiment. Also, the concentrations of SPs used are missing. In this part it says that the effect of SPs was investigated, but only SP6 and SP10 were examined, like in all other assays. It is necessary to harmonize the writing of methods. It can be written at the beginning of methods description that based on the results of bacterial growth assay only SP6 and SP10 were chosen for further investigation. Also, is it possible to present the results of biofilm assay as a % of biofilm biomass, that way you have a clearer picture of results obtain, and you can discuss about inhibition of biofilm formation by SPs.

We have revised the text based on the reviewer’s suggestion about concentrations of SP solutions (SP6, SP10; 100, 300, 500, and 1,000 μg/mL) and the number of CFU/ml of S. mutans used in biofilm forming assay. In addition, the reason why we focused on SP6 and SP10 for biofilm forming assay and other experiments, we revised the text in bacterial growth assay.

Bacterial iron uptake assay subsection: give information about concentrations of SP used. Please explain way was the iron uptake followed for 1h and 18h.

We have revised the text based on the reviewer’s suggestion about concentrations of SP solutions (SP6 and SP10; 100 and 1,000 μg/mL) used in bacterial iron uptake assay. To determine whether SPs affect iron uptake in the early (1h) or late (18h) growth phase of S. mutans, we performed the assay over this time course.

Bacterial morphological observation subsection: give information about SPs concentration used.

We have revised the text based on the reviewer’s suggestion about concentrations of SP solutions (SP6 and SP10; 1,000 μg/mL) used in bacterial morphological observation.

Bacterial viability analysis subsection: This whole part of methodology in not clearly written. Reformulate in order to make sense.

We have revised the text in bacterial viability analysis.

RESULTS
Presentation of results is not good. Please explain why was only chitosan investigated in first experiment (inhibition of bacterial growth). If you wanted to test basic formulas of SPs, then why did you test histidine and glucosamine as well, and in all experiments.

Thank you for your suggestion. Indeed, if the basic formula of SP has no antimicrobial activity, we had better to provide data on glucosamine and histidine in addition to chitosan. In this case, since chitosan has no antimicrobial activity, and since the literature search did not reveal any papers on the antimicrobial activity of glucosamine and histidine, we believe that the three molecules used as the basic formula of SPs have no antimicrobial activity. In order not to confuse the reader, we remove the data of chitosan from Figure 2.

Is it necessary to test H2O? Even though SPs are dissolved in water, in my opinion it is not necessary to test it as solvent control. In case you want water to stay, then all statistic analysis must be done compared to solvent control not to no drug. And solvent control must be compared in that case to no drug.

As the reviewer pointed out, we don't think it is necessary to present the data of water used as a solvent; we remove the H2O condition from the other data set.

The concentration selection for testing bacterial iron uptake as well as viability is questionable. You stated that concentration of 1000 μg/ml was used, but at that concentration based on the results of bacterial growth inhibition assay no growth was detected. So, if there are no bacteria it is not possible to test effect of SPs on their iron uptake. Furthermore, how did you test the ability of SPs to chelate iron? Whole research is based on the ability of these compounds to competitively bind iron ions and consequently reduce ability of bacteria to chelate iron that can lead to their death.

In this study, we think that SP6 and SP10 have bacteriostatic effects (inhibition of bacteria growth) rather than bactericidal effects on S. mutans. Therefore, we believe that the measurement is possible because S. mutans is not killed by the action of 1,000 μg/ml of SPs as in the iron uptake experiment. In addition, co-author Ohara examined SPs basic chelating abilities using standard ferric (Fe3+) water solution by the sulfosalicylic acid visual colorimetric method. SP6 and SP10 chelated Fe3+ in a dose- dependent manner. A dose of 0.5 mg/mL SP6 chelated more than 75% of iron, and 0.1 mg/mL SP10 chelated more than 80% of iron (ref. 22).

It would be good to present results of cytotoxicity as % of cell survival, not as absorbance.

Thank you for your suggestion. We have revised the Figure 7 following the reviewer’s suggestion.

DISCUSSION
Discussion must be more detailed, explaining obtained results. In present form, in discussion part, results are just repeated, without any specific explanation. This whole part must be expanded and written again.

We have revised the text in discussion section.

Reviewer 2 Report

Comments and Suggestions for Authors

The manuscript " Novel iron chelators, super-polyphenols, show antimicrobial effects against cariogenic Streptococcus mutans " investigates the inhibitory effects of novel iron chelators, the super-polyphenols (SPs), on Streptococcus mutans, in vitro.

The manuscript is well organized in its paragraph division, simplifying reading especially in the "Results" and "Materials and Methods" sections.

The Discussion section is lacking and requires further editing and rewriting.

Below are some suggestions and doubts for each section of the manuscript:

Introduction:

-        The two sentences "As a result, the growing elderly population in Japan has consequently led to an increase in the number of patients who are unable to clean their teeth without assistance. Unfortunately, mechanical removal such as brushing and scaling is still the main method for removing dental biofilm and maintaining oral hygiene." need at least one reference.

-        The term "oral biofilm infection" may be confusing and misleading. Oral infections are due to microorganisms, not oral biofilm. The oral biofilm is a complex unit not only composed of microorganisms.

-        The introduction starts with the definition of dental caries. The etiology of caries includes Streptococcus mutans. Next, the link between Porphyromonas gingivalis, a periodontopathogenic bacterium, and iron is explained. However, there is no mention of Streptococcus mutans in the introduction, neither in relation to caries nor the link with iron. It would be appropriate to include a section for Streptococcus mutans in the introduction, as it is the bacterium whose sensitivity to SPs has been investigated.

-        The purpose of the study written at the end of the introduction and in the discussion is not in line with what is reported in the abstract. The aim of the study is to evaluate the efficacy of SPs on S.mutans, not on oral infections, which are numerous. Rewrite the aim of the study uniformly between the abstract, introduction, and discussion. Choose a sentence that is as precise as possible and less generic, so that it specifically explains the purpose of this manuscript.

Materials and Methods:

-        The abbreviations "CFU", "prm", "P" (of the p-value), were written without the abbreviation in writing beforehand.

Results:

-        I am not clear on what you mean by "S.mutans biofilm formation". Can you explain it?

Discussion:

-        The full stop is misplaced in the first sentence.

-        The sentence " From this perspective, the medical world is faced with the need to control bacterial infections with novel approaches, and oral infections associated with oral biofilms are no exception." suggests that oral biofilm is the cause of oral infections. The introductions also included this concept. Oral biofilm does not cause infections, but it is microorganisms in a dysbiosis condition that bring infection. S.mutans, as well as other microorganisms, can be present in the oral cavity even under healthy conditions. Check and rewrite these important concepts throughout the manuscript in a less misleading manner.

-        After reference 23, the "Discussion" section only presents the results of the study, but there is no real discussion of them. Only one hint of discussion reappears towards the end with the reference to Ohara's study.

Discuss the results obtained from the present study. Are there any previous similar studies in the literature, including on other microorganisms or in other body districts? Were the results obtained expected? Etc.

-        The limitations of the study are indicated, but what are the strengths?

Conclusions:

-        The Conclusion section is missing. However, in my opinion, it could be added separately to the discussion, providing the reader with a single short sentence on the result obtained for each paragraph of the results.

-        In the discussion, comment on the results obtained. In the conclusion, provide a brief summary of the results. The last 4 lines of the discussion could also be moved to the conclusion section.

References:

-        References in the text should be given in square brackets, not round. Also delete spaces in brackets between consecutive references.

-        References in the "References" section are not formatted according to the format required by the journal. Please check and reformat all references.

-        Move reference 18 after the name "Ohara et al." (introduction section) and not at the end of the sentence.

-        Many references are very old. One dates as far back as 1944 and many between 1970-1990. It would be appropriate, where possible, to include more recent articles and evidence.

Author Response

Reviewer 2
The manuscript " Novel iron chelators, super-polyphenols, show antimicrobial effects against cariogenic Streptococcus mutans " investigates the inhibitory effects of novel iron chelators, the super-polyphenols (SPs), on Streptococcus mutans, in vitro.
The manuscript is well organized in its paragraph division, simplifying reading especially in the "Results" and "Materials and Methods" sections.
The Discussion section is lacking and requires further editing and rewriting.

Thank you for reviewing the manuscript and providing several useful comments. We have provided point- by-point responses to each comment, and the red text indicates the revised manuscript. In addition, we have revised the text in discussion section.

Below are some suggestions and doubts for each section of the manuscript:
Introduction:
-The two sentences "As a result, the growing elderly population in Japan has consequently led to an increase in the number of patients who are unable to clean their teeth without assistance. Unfortunately, mechanical removal such as brushing and scaling is still the main method for removing dental biofilm and maintaining oral hygiene." need at least one reference.

Thank you for your suggestion, we have revised the text and add the reference (ref.6).

-The term "oral biofilm infection" may be confusing and misleading. Oral infections are due to microorganisms, not oral biofilm. The oral biofilm is a complex unit not only composed of microorganisms.

Thank you for your suggestion, we have revised the term “oral biofilm infection”.

-The introduction starts with the definition of dental caries. The etiology of caries includes Streptococcus mutans. Next, the link between Porphyromonas gingivalis, a periodontopathogenic bacterium, and iron is explained. However, there is no mention of Streptococcus mutans in the introduction, neither in relation to caries nor the link with iron. It would be appropriate to include a section for Streptococcus mutans in the introduction, as it is the bacterium whose sensitivity to SPs has been investigated.

Thank you for your suggestion, we have revised the introduction section including S. mutans.

-The purpose of the study written at the end of the introduction and in the discussion is not in line with what is reported in the abstract. The aim of the study is to evaluate the efficacy of SPs on S.mutans, not on oral infections, which are numerous. Rewrite the aim of the study uniformly between the abstract, introduction, and discussion. Choose a sentence that is as precise as possible and less generic, so that it specifically explains the purpose of this manuscript.

Thank you for your suggestion, we have revised the aim of this study following the reviewer’s suggestion “to evaluate the efficacy of SPs on S. mutans”.

Materials and Methods:
-The abbreviations "CFU", "prm", "P" (of the p-value), were written without the abbreviation in writing beforehand.

We have added the abbreviation lists of each word.

Results:
-I am not clear on what you mean by "S.mutans biofilm formation". Can you explain it?

Thank you for your suggestion, we have revised the term. And we have added the reason why we investigated the effect of SPs on the biofilm formation of S. mutans in the result section.

Discussion:
- The full stop is misplaced in the first sentence.

Thank you for your suggestion, we have revised the full stop position.

-The sentence " From this perspective, the medical world is faced with the need to control bacterial infections with novel approaches, and oral infections associated with oral biofilms are no exception." suggests that oral biofilm is the cause of oral infections. The introductions also included this concept. Oral biofilm does not cause infections, but it is microorganisms in a dysbiosis condition that bring infection. S.mutans, as well as other microorganisms, can be present in the oral cavity even under healthy conditions. Check and rewrite these important concepts throughout the manuscript in a less misleading manner.

Thank you for your suggestion, we have revised the sentence.

- After reference 23, the "Discussion" section only presents the results of the study, but there is no real discussion of them. Only one hint of discussion reappears towards the end with the reference to Ohara's study.

Thank you for your suggestion, we have revised the text in discussion section.

Discuss the results obtained from the present study. Are there any previous similar studies in the literature, including on other microorganisms or in other body districts? Were the results obtained expected? Etc.

Thank you for your suggestion, we have added reference (ref. 27) about the recent research the disease- alleviating effects of specific iron binding by oral DFO application in the infection of Campylobacter jejuni induced acute enterocolitis in mouse model in discussion section.

-The limitations of the study are indicated, but what are the strengths?

Thank you for your suggestion, we have revised the text in conclusion section including the strength of this study. In this study, we believe that the high biosafety of SP6 and SP10 compared to commercially available iron chelators, including DFO and DFX, and SPs have a bacteriostatic (growth inhibiting) rather than bactericidal effect on S. mutans, which means the possibility of inhibiting the development of AMR.

Conclusions:
-The Conclusion section is missing. However, in my opinion, it could be added separately to the discussion, providing the reader with a single short sentence on the result obtained for each paragraph of the results.

Thank you for your suggestion, we have revised the manuscript and added a conclusion section. In this section, we have briefly summarized the results of this study and mentioned future possibilities.

-In the discussion, comment on the results obtained. In the conclusion, provide a brief summary of the results. The last 4 lines of the discussion could also be moved to the conclusion section.

Thank you for your suggestion, we have revised the text and the 4 line so the discussion move to the conclusion section.

References:
-References in the text should be given in square brackets, not round. Also delete spaces in brackets between consecutive references.

-References in the "References" section are not formatted according to the format required by the journal. Please check and reformat all references.
-Move reference 18 after the name "Ohara et al." (introduction section) and not at the end of the sentence.

Thank you for your suggestion, we followed Journal instruction.

-Many references are very old. One dates as far back as 1944 and many between 1970-1990. It would be appropriate, where possible, to include more recent articles and evidence.

Some old reference is important for discussing the relationship of iron and bacteria. But, we have added the recent research related to iron chelate therapy’s side effect in discussion section.

Reviewer 3 Report

Comments and Suggestions for Authors

 General comments

 This manuscript evaluates the effectiveness of novel iron chelators to control antimicrobial cariogenic Streptococcus mutans.  The study is of interest to the field of dental care. The experimental work and the analysis of data are performed well providing useful and new information. Moreover, results are clearly commented.  However, a few changes and  clarifications are necessary before the manuscript can be accepted for publication.

 Specific comments

 Abstract

Replace bacteria by bacterium at the fourth line.

Add the info about the chemical composition of SP6 and SP10 as it is mentioned in supplementary material.

 Materials and methods

why the authors used logarithmic growth phase since at this stage bacteria are more sensitive to stress factors?

 Results and discussion

Add information about the toxicological evaluation of SP6 and SP10.

Comments on the Quality of English Language

Revise the language. Avoid the use of we, writing must be impersonal.

Author Response

Reviewer 3

General comments
This manuscript evaluates the effectiveness of novel iron chelators to control antimicrobial cariogenic Streptococcus mutans. The study is of interest to the field of dental care. The experimental work and the analysis of data are performed well providing useful and new information. Moreover, results are clearly commented. However, a few changes and clarifications are necessary before the manuscript can be accepted for publication.

Thank you for reviewing the manuscript and providing several useful comments. We have provided point- by-point responses to each comment, and the red text indicates the revised manuscript.

Specific comments
Abstract
Replace bacteria by bacterium at the fourth line.

Thank you for your suggestion, we have revised the term following the reviewer’s suggestion.

Add the info about the chemical composition of SP6 and SP10 as it is mentioned in supplementary material.

We have revised the text based on the reviewer’s suggestion in materials and methods section related to SPs.

Materials and methods
Why the authors used logarithmic growth phase since at this stage bacteria are more sensitive to stress factors?

As the reviewer pointed out, we used bacteria in the logarithmic growth phase because we wanted to observe whether SP6 and SP10 were really effective on the bacteria, since bacteria are more sensitive to stress when they are in the growth phase.

Results and discussion
Add information about the toxicological evaluation of SP6 and SP10.

Co-author Ohara has examined acute toxicity tests in rats were performed to evaluate the basic toxicity of SP6 and SP10. SPs were orally administered (600 and 1,000 mg/kg), and the body weight of the rats did not change compared to the control group during the examination. No abnormal behaviors were observed through the period of examination. A blood test was also performed in rats treated with SP6 (1,000 mg/kg) and SP10 (1,000 mg/kg), and no significant abnormalities were observed. To evaluate any possible adverse effects to the organs, Ohara examined the liver and kidney tissues and found no injuries in the specimens. To compare the safety of SP6 and SP10 to the known iron chelator DFO, an intravenous injection test was performed. Although all mice died immediately after intravenous administration of DFO (300 mg/kg), none died after administration of SP6 or SP10 (300 mg/kg) (ref. 22). We think these results demonstrated that SP6 and SP10 are basically safer than DFO.

Comments on the Quality of English Language
Revise the language. Avoid the use of we, writing must be impersonal.

Thank you for your suggestion. We have revised the manuscript following the reviewer’s suggestion.

Reviewer 4 Report

Comments and Suggestions for Authors

In this manuscript titled “Novel iron chelators, super-polyphenols, show antimicrobial effects against cariogenic Streptococcus mutans” Shinoda-Ito et al investigated invitro efficacy of super poly-phenols which are iron chelators against Streptococcus mutans, causative agent of dental caries. This study is intriguing and novel. Nevertheless, some minor revisions are requisite prior to its publication, and the following are my remarks:

1.      There are few grammatical mistakes in the manuscript (highlighted in red and yellow colors), for example:

a.      Abstract: “SPs were developed to reduce the side effects on of iron chelation therapy and were either water soluble or insoluble depending on their isoforms.”

b.      Discussion: “Specifically, SP 6 SP6 and SP10 have the same chelating structure, with three hydroxy groups on benzene, and has have been shown to inhibit bacterial growth of S. mutans”

2.      In bar graphs containing H2O should be corrected as H2O.

3.      Discussing about super poly-phenols and their properties including anticancer, antibacterial and antiviral activity will help reader to know the importance of SPs.

4.      The authors have not provided accurate referencing for example, ref#19 is missing from the manuscript. Additionally, authors should refer to new reports for increased antimicrobial resistance and novel studies on super poly-phenols or deferasirox.

Author Response

Reviewer 4
In this manuscript titled “Novel iron chelators, super-polyphenols, show antimicrobial effects against cariogenic Streptococcus mutans” Shinoda-Ito et al investigated invitro efficacy of super polyphenols which are iron chelators against Streptococcus mutans, causative agent of dental caries. This study is intriguing and novel. Nevertheless, some minor revisions are requisite prior to its publication, and the following are my remarks:

Thank you for reviewing the manuscript and providing several useful comments. We have provided point- by-point responses to each comment, and the red text indicates the revised manuscript.

1. There are few grammatical mistakes in the manuscript (highlighted in red and yellow colors), for example:
a. Abstract: “SPs were developed to reduce the side effects on of iron chelation therapy and were either water soluble or insoluble depending on their isoforms.”

b. Discussion: “Specifically, SP 6 SP6 and SP10 have the same chelating structure, with three hydroxy groups on benzene, and has have been shown to inhibit bacterial growth of S. mutans”

Thank you for your suggestion, we have revised the text following the reviewer’s suggestions.

2. In bar graphs containing H2O should be corrected as H2O.

Thank you for your suggestion, we have revised the figure following the reviewer’s suggestions.

3. Discussing about super polyphenols and their properties including anticancer, antibacterial and antiviral activity will help reader to know the importance of SPs.

Thank you for your suggestion, we have revised the discussion section following the reviewer’s suggestions except antiviral activity because we have not checked the antiviral activity of SPs.

4. The authors have not provided accurate referencing for example, ref#19 is missing from the manuscript. Additionally, authors should refer to new reports for increased antimicrobial resistance and novel studies on super polyphenols or deferasirox.

Thank you for your suggestion, we have checked the references and added the appropriate reference (ref. 27).

Round 2

Reviewer 1 Report

Comments and Suggestions for Authors

After I reviewed the modified version of the manuscript I can confirm that  the authors have accepted the proposed changes, therefore I recommend this paper for the publication. 

Reviewer 2 Report

Comments and Suggestions for Authors

The manuscript has been extensively revised and reorganized in all its sections.

I congratulate the authors for improving each section.